# Embodied Crowd Counting

**Runling Long**[1], **Yunlong Wang**[1], **Jia Wan**[1],[*] **Xiang Deng**[1], **Xingting Zhu**[2], **Weili Guan**[1],
**Antoni B. Chan**[2], **Liqiang Nie**[1]
[1]Harbin Institute of Technology, Shenzhen
[2]City University of Hong Kong
{24b951033,220110608}@stu.hit.edu.cn
{jiawan1998,nieliqiang}@gmail.com
{dengxiang,guanweili}@hit.edu.cn
{xt.zhu}@my.cityu.edu.hk
{abchan}@cityu.edu.hk

## Abstract

Occlusion is one of the fundamental challenges in crowd counting. In the community, various data-driven approaches have been developed to address this issue, yet their effectiveness is limited. This is mainly because most existing crowd counting datasets on which the methods are trained are based on passive cameras, restricting their ability to fully sense the environment. Recently, embodied navigation methods have shown significant potential in precise object detection in interactive scenes. These methods incorporate active camera settings, holding promise in addressing the fundamental issues in crowd counting. However, most existing methods are designed for indoor navigation, showing unknown performance in analyzing complex object distribution in large-scale scenes, such as crowds. Besides, most existing embodied navigation datasets are indoor scenes with limited scale and object quantity, preventing them from being introduced into dense crowd analysis. Based on this, a novel task, Embodied Crowd Counting (ECC), is proposed to count the number of persons in a large-scale scene actively. We then build up an interactive simulator, the Embodied Crowd Counting Dataset (ECCD), which enables large-scale scenes and large object quantities. A prior probability distribution approximating a realistic crowd distribution is introduced to generate crowds. Then, a zero-shot navigation method (ZECC) is proposed as a baseline. This method contains an MLLM-driven coarse-to-fine navigation mechanism, enabling active Z-axis exploration, and a normal-line-based crowd distribution analysis method for fine counting. Experimental results show that the proposed method achieves the best trade-off between counting accuracy and navigation cost. Code can be found at https://github.com/longrunling/ECC?.

## 1 Introduction

Crowd counting is critical for public safety and urban planning [23]. One main challenge in this field is occlusion. It can be categorized into two aspects: overlap and invisibility. Overlap refers to the high density of people stacked together, making it difficult to distinguish each individual from some viewpoints, while invisibility indicates that the current camera position is obstructed, such as being blocked by buildings, or far from the crowds so that the target cannot be detected clearly. To summarize, these situations are caused by a poor observation point. Existing methods try to solve these challenges from several perspectives, such as using multi-scale feature extraction [22], body part detection [30], or multi-camera fusion [56, 57]. Datasets are also expanded for enhancing method

---

[*]Corresponding author

39th Conference on Neural Information Processing Systems (NeurIPS 2025).

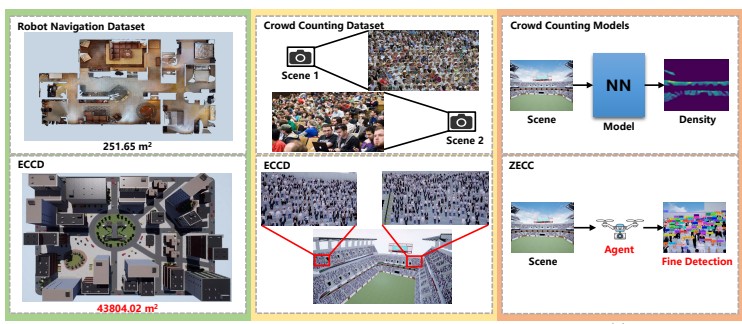

(a)              (b)              (c)

Figure 1: (a) Comparison between ECCD and existing embodied navigation datasets. ECCD features large-scale outdoor crowd scenes. (b) Comparison between ECCD and crowd counting datasets. ECCD enables interactive ability. (c) Comparison between ZECC and existing crowd counting methods. ZECC is an agentic framework with automatic camera adjusting ability.

Table 1: Comparison between ECCD and other related datasets. ECCD combines features from both crowd counting datasets and embodied navigation datasets.

| Dataset | Active camera | Place | Max target quantity / sample | Dynamic target | w/o Instruction | Task |
|---|---|---|---|---|---|---|
| NWPU-Crowd [50] | ✗ | - | 20,033 | ✗ | - | CC |
| SenseCrowd [26] | ✗ | - | 296 | ✓ | - | CC |
| DukeMTMC [40] | ✗ | - | 2,834 | ✓ | - | CC |
| R2R [1] | ✓ | Indoor (Ground) | - | ✗ | ✗ | VLN |
| HM3DSem [38] | ✓ | Indoor (Ground) | - | - | ✓ | ObjNav |
| AerialVLN [12] | ✓ | City(Aerial) | - | ✗ | ✗ | VLN |
| CityNav [25] | ✓ | City(Aerial) | - | ✗ | ✗ | VLN |
| Openfly [11] | ✓ | City(Aerial) | - | ✗ | ✗ | VLN |
| ECCD | ✓ | City(Aerial) | 15,488 | ✗ | ✓ | ECC |

performance [58, 21, 42, 50]. In general, these methods or datasets either try to enhance learned model representation or expand the camera sensing range by introducing multi-view settings. Yet, their passive camera settings do not fully solve the occlusion challenge in crowd counting, especially in cases where crowds exceed the sensing range, or no cameras are set to detect completely obscured crowds. These issues restrict passive-camera-based methods in practice.

Recent development in Embodied AI brings a new perspective to address the occlusion challenge in crowd counting. It has been demonstrated to possess significant potential in enhancing scene exploration and object detection. Aspects such as Vision-Language-Navigation (VLN) have been directed towards equipping mobile robots with human-like perception abilities, resulting in remarkable performance in exploring scenes and detecting objects in open environments [9, 5, 59, 8]. The active camera settings in VLN are promising to solve the fundamental challenges in crowd counting, since it can optimize observation points to mitigate overlap and invisibility caused by fixed camera position settings. Yet most VLN benchmarks [51, 46, 4, 2] are indoor environments with limited exploration space (e.g., no Z-axis action options) and relatively small object quantity. And the performance of such methods remains unknown for detecting crowds with a large quantity and complex distribution in large-scale scenes. This results in a significant gap between VLN and crowd counting, as in practice, crowds often appear in large spaces with varying distribution.

To address the issues, we first define a novel task, Embodied Crowd Counting (ECC), which is shown in Figure 1. The task is defined as counting the total number of people present in a large outdoor scene using a drone. Given the absence of an existing dataset, we have created a new dataset called the Embodied Crowd Counting Dataset (ECCD) specifically for this task. This dataset includes 60 unique and diverse large-scale outdoor scenes. Each scene spans an area of up to 40,000 $m^2$ and has a target crowd size of up to 15,000 individuals. To ensure realism in our dataset, we employ a Poisson Point Process [13] for modeling the distribution of crowd sizes, which effectively simulates real-world crowd scenarios. The comparision of ECCD and related datasets are shown in Table 1.

In this study, we introduce a baseline method, Zero-shot Embodied Crowd Counting (ZECC) aiming at counting individuals present in environments populated by crowds. Given the ability of interacting dynamically with surroundings using foundation models, modern argentic methods achieve even better performance against training-based methods while maintaining generalization ability. [9, 24, 12, 31]. Inspired by this, we aim to build a zero-shot baseline that can generate to diverse scenes leveraging foundation models, using an argentic paradigm without a vast amount of training data. The primary challenge lies in choosing suitable navigation points to find a balance between efficient exploration

and effective detection. To address this, our approach consists of two main components: the Active Top-down Exploration (ATE) method and the Normal-line Based Navigation (NLBN) system. The ATE method serves as an efficient exploration strategy that employs a coarse-to-fine navigation approach. It utilizes the common sense capabilities of Multi-Modal Large Language Models (MLLM) to assess the environment, allowing for effective planning of vertical movement (Z-axis exploration). This takes advantage of the six degrees of freedom (6-DoF), which helps to avoid obstacles at lower altitudes and provides a broader view of the surroundings, thereby facilitating better exploration. Following the rough estimation of crowd distribution produced by ATE, we propose the NLBN to create precise navigation points for improved crowd detection. By using the normal lines of surfaces, we can establish detailed observation points that achieve a balanced trade-off between exploration efficiency and detection performance. This method addresses challenges related to overlapping individuals and visibility issues, ultimately leading to more accurate crowd counting. Experimental results show that the method is effective due to its interactive ability. The contributions of this work can be summarized as follows:

- We present an innovative task called ECC, specifically designed to address the challenges of occlusion and multi-scale complexities that are prevalent in conventional crowd counting methods.

- A new dataset called ECCD has been collected to redefine the landscape of crowd analysis. Unlike traditional crowd counting and VLN datasets, it features large-scale outdoor crowd scenes with interactive capabilities.

- We propose a baseline method, ZECC, using MLLM for Z-axis exploration, reducing costs while ensuring detection performance. By utilizing normal lines to calculate navigation points, this approach eliminates occlusion and enhances visibility in crowds.

## 2 Related works

### 2.1 Crowd Counting

Crowd-counting algorithms have greatly benefited from large-scale, high-quality datasets like UCF-CC50 [20] and UCF-QNRF [21]. These foundational datasets have facilitated the creation of subsequent collections focused on dense crowd imagery, such as ShanghaiTech [58], JHU-CROWD++ [42], NWPU-Crowd [50]. However, the images in these datasets are generated from fixed cameras. In contrast, ECCD provides interactive capabilities while maintaining diverse crowd distribution.

Methods like [58, 22, 27, 6] leverage crowd distribution prior in an image, or use attention maps to learn dependency. Recent multi-modal approaches [10, 54, 33, 48] leverage vision-language models to transfer image-text knowledge to dense crowd prediction. While these models improve long-range and overlapping small target detection ability, their performance is restricted if the overlap reaches an extreme level. Recent efforts expand spatial coverage by multi-view systems [37, 19] that use multiple cameras to capture images from large-scale scenes. Others like [17, 16] use recorded video to conduct crowd analysis. Although these advances represent significant progress in addressing the basic challenges of crowd counting compared to traditional methods that rely on fixed images, the camera settings remain predetermined by the dataset. This restriction means that the settings cannot be modified during the inference process, limiting the ability to fully examine larger environments. ZECC allows active exploration, which is fundamentally different from existing methods.

### 2.2 Embodied Navigation

Many traditional robot navigation methods were developed using conventional navigation datasets like KITTI [14] and SUN RGB-D [44]. Beyond these foundational datasets, [51, 38] provide 3D indoor environments for navigation and interaction tasks. These datasets are limited with their small scale size and small object quantity. Recent datasets have been created towards outdoor navigation [29, 25, 11, 28] However, they are designed for VLN tasks without consideration of object quantity. Compared to these datasets, ECCD simultaneously supports large-scale outdoor scenes and large object quantity with diverse distribution.

Efficient exploration using mobile robots remains a crucial challenge in vision and robotics. [15, 18, 35, 44, 12] have developed human-like cognitive maps, enabling autonomous path learning in unknown environments. Other approaches like [3, 7, 36] use reinforcement learning to develop

exploration policies. Recently, [55] applied video-based visual-language models to plan sequential actions in VLN. And [31, 60] presented zero-shot models that utilize natural language instructions to guide agents through environments without prior environment-specific training. However, the methods mentioned above are for indoor navigation without Z-axis moving ability. This restricts their ability in large-scale outdoor scenes. Recently, methods like [29, 52] introduce 6-DoF in outdoor VLN. Yet they are designed for instruction following tasks, in which the movement is restricted by human language. Besides, they lack the ability to analyze crowds with a large quantity and complex distribution. In contrast, ZECC is the first self-motivated agent in 6-DoF that can handle crowds.

## 3 Method

### 3.1 Problem Definition

To ensure the interactive with environment for accurately crowd counting in vast outdoor environments, we propose an innovative task called Embodied Crowd Counting (ECC). In ECC, an agent is first deployed in an unknown environment. At time step $t$, the input of the agent is RGB-D observations $O_t$ along with its pose $p_t$. Based on this information, the agent predicts navigation point $p \subset \mathbb{R}^3$ to a drone at time step $t$ and the drone moves to $p$. During exploration, the agent is allowed to record observations that help with crowd counting. When the agent decides to stop, it outputs an integer that represents its crowd counting result. The counting error and travel distance are calculated to assess the agent's performance. ECC can be considered analogous to the Zero-Shot Object-Goal Navigation (ZSON) task [32], since they both require an agent to detect targets in an unexplored environment without additional assistance. However, ECC faces unique challenges: 1) The Z-axis is available, which is not considered by most ZSON methods. 2) Complex crowd distributions exist, including heavy occlusions, while ZSON methods only consider fewer objects. Current ZSON methods show unknown performance under such differences, and new methods should be considered in ECC. Note that ECC is different from the instruction following tasks in VLN such as [29, 52], since these tasks require natural language assistance to drive the agent.

### 3.2 Embodied Crowd Counting Dataset (ECCD)

We propose a new dataset called ECCD to support algorithm design and evaluation for the community, developed using Unreal Engine 4. This platform enables programmable environment construction, allowing scene richness and scalability. The characteristics of ECCD are as follows:

**Diversity.** Since different environments have different layouts, crowd distributions, and quantities reflect different challenges, ECCD is designed to contain diverse scenes. This dataset contains 60 distinct environments. Also, ECCD has an area up to 40,000 $m^2$, reaches a height up to 50 $m$, and allows for the simultaneous presence of over 15,000 targets within a single scene. This is significantly different from existing robot navigation datasets, such as [38], which offers environments with an average navigable space of 1,000 $m^2$, and crowd counting datasets, such as [58], which contains 1,198 annotated images captured with static cameras and a total of 330,165 individuals.

**Realism.** The ECCD is designed to simulate large-scale outdoor crowd scenarios in reality, in order to ensure practical effectiveness of systems built on ECCD, as shown in Figure 2 (a). Environments like a city, a stadium, and a parking lot are simulated. Additionally, the environments support a complex structure of buildings in real life, such as multiple floors and bleachers. These features allow ECCD to reflect challenges in reality.

**Crowd generation mechanism.** To model real distribution in crowd counting, ECCD uses Poisson Point Process for crowd quantity distribution modeling [13]. In ECCD, Blocks are placed in potential areas where crowds may exist. Then, for each block $\mathcal{U} \subset \mathbb{R}^2$, the process is defined as:

$$\mathbb{P}\left(N(\mathcal{U}) = k\right) = \frac{(\lambda \cdot |\mathcal{U}|)^k}{k!} e^{-\lambda \cdot |\mathcal{U}|}, \quad k \in \mathbb{N}, \tag{1}$$

where $N(\mathcal{U})$ denotes the number of individuals in block $\mathcal{U}$, $\lambda$ represents the crowd density set by human experts according to the environment semantics, and $|\mathcal{U}|$ is the area of the block. This ensures ECCD generates crowds that approximate real situations. By comparison, existing simulators based on UE4, such as AirVLN and OpenUAV, do not consider object quantity and distribution.

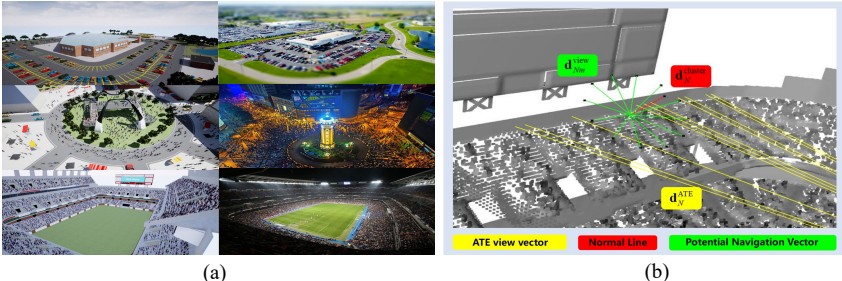
(a)

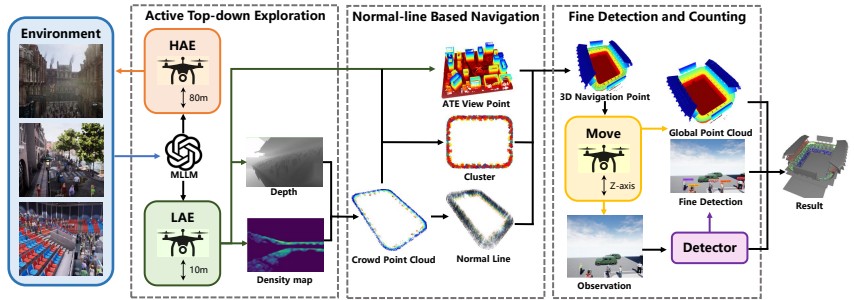
(b)

Figure 2: (a) ECCD is designed to mimic building and crowd distribution realistically. On the left are samples from ECCD, and on the right are the real scenes. (b) Illustration of the potential navigation vectors, normal lines, and FBE view vectors. Zoom in for better visualization.

Figure 3: The proposed framework. First, ATE is proposed to estimate the global crowd distribution efficiently. Then, NLBN is proposed to generate fine observation points, alleviating crowd overlap. The final result is generated by aggregating all fine detections.

### 3.3 Zero-shot Embodied Crowd Counting (ZECC)

#### 3.3.1 Overview

Previous embodied navigation agents are designed for indoor environments [31], or relying on language assistance [29, 52], making them restricted in large-scale outdoor scenes. Under such a context, a zero-shot agent, ZECC, which can actively control altitude and conduct crowd analysis, is proposed. As illustrated in Figure 3, the method consists of three components: Active Top-down Exploration (ATE), Normal-line based Navigation (NLBN), and Fine detection. ATE is designed for adjusting agent altitude for efficient coarse crowd distribution estimation, and NLBN estimates normal lines on top of the crowd for accurate crowd observation, alleviating occlusion.

#### 3.3.2 Active Top-down Exploration (ATE)

Crowds are often concentrated in specific areas, such as roads and squares, making it unnecessary to explore every region of the environment. To address this, ATE is proposed to estimate global crowd distribution by changing altitude. This approach aims to improve efficiency by focusing exploration efforts on the most relevant areas. In particular, high-altitude exploration (HAE) brings a broader field of view for decision making, as well as relatively less exploration cost since obstacles are often sparse in high altitude. In contrast, low-altitude exploration (LAE) provides close-range observation for precise target detection. The agent needs to plan HAE and LAE to achieve efficiency and accuracy simultaneously. Therefore, ATE leverages the Z-axis mobility of outdoor agents and the common sense of MLLM to switch between HAE and LAE using local environment layout reasoning. Then, the crowd distribution is estimated by a crowd counting model to predict density maps.

Specifically, the agent collects observations $O_t = \{o_t^1, \ldots, o_t^c\}$ and pose $p_t$ at time step $t$, where $c$ indicates the camera number of the agent. Then, an MLLM is prompted using observations and text prompts to conduct environment layout reasoning. The MLLM is asked to predict whether the current location is valuable for LAE, by referring to the current crowd appearance and obstacle layout. This process is formulated as:

$$s_t = \text{MLLM}\left(O_t; I\right), \tag{2}$$

where MLLM $(\cdot)$ is the inference process and $I$ is the prompt. $s_t \in [0, 1]$. If $s_t > 0.5$, the agent will adjust its altitude for LAE. After switching the altitude strategy, the agent will conduct regular exploration. For HAE, one of the frontiers between explored and unknown areas is selected as a navigation point. For LAE, the agent keeps exploring until areas within its field of view during HAE are fully explored. During LAE, once the agent reaches a navigation site $f$, it gathers observations $O_f = \left\{ o_f^1, \ldots, o_f^c \right\}$, and a crowd counting model predicts crowd density maps on the observations. Then, the density maps are projected onto the global point cloud to form a global crowd distribution:

$$d_f = \mathrm{P}\left(\mathrm{G}\left(O_f\right), p_f\right), \tag{3}$$

where $\mathrm{P}\left(\cdot\right)$ is the projection operation, $\mathrm{G}\left(\cdot\right)$ is a crowd counting model, and $p_f$ is the agent pose at $f$.

### 3.3.3   Normal-line based Navigation (NLBN)

NLBN is designed to analyze overlapping structures in dense crowds by converting the crowd detection task into surface detection, which helps identify optimal observation points. While random viewpoints may impair visibility, vantage points located above the center of the crowd provide clearer views for individual identification. This top-down perspective enables the distinction of overlapping individuals and maintains targets within the field of view (FOV). Initially, NLBN clusters the crowd point cloud into subregions using a clustering method, simplifying the analysis by breaking the large point cloud into manageable parts, as navigating to each point can be resource-intensive. Surfaces are then fitted to derive normal lines, thus transforming complex crowd analysis into a more straightforward surface analysis. Finally, optimized navigation points are generated based on these normal lines, ensuring they are elevated to facilitate accurate crowd detection while employing a viewpoint-based approach to avoid overlaps with obstacles.

In particular, GMM [41] is used to divide the global crowd distribution into manageable subregions. It can divide any crowd distribution into patches. A parameter $\epsilon$ is used to determine the size of each GMM cluster. Surfaces are fitted for each patch afterwards. Then, for the $N$-th cluster, the normal is obtained and represented by $\mathbf{d}_N^{\mathrm{cluster}}$. Candidate view directions $\{\mathbf{d}_{N1}^{\mathrm{view}}, ..., \mathbf{d}_{Nm}^{\mathrm{view}}\}_N$ are sampled under angular constraints:

$$\frac{\mathbf{d}_N^{\mathrm{cluster}} \cdot \mathbf{d}_{Nm}^{\mathrm{view}}}{\|\mathbf{d}_N^{\mathrm{cluster}}\|\|\mathbf{d}_{Nm}^{\mathrm{view}}\|} = \zeta, \tag{4}$$

where $\zeta$ is a hyper parameter. These candidate view directions can bring the agent to optimized observation points by selecting a position along the vector. However, there are two issues: 1) The position may be located on obstacles; 2) The crowd cluster may be out of the agent's FOV. Since the propagation of light naturally points to unobstructed areas, the ATE viewpoints are used to generate the final navigation points, which are calculated as:

$$\mathbf{d}_N^{\mathrm{ATE}} = \mathbf{x}_N^{\mathrm{ATE}} - \mathbf{x}_N^{\mathrm{cluster}}, \tag{5}$$

where $\mathbf{x}_N^{\mathrm{ATE}}$ is the navigation point from which the agent finds the cluster center $\mathbf{x}_N^{\mathrm{cluster}}$ during ATE. Then, the potential navigation directions that are at the minimum angles to the ATE view vectors are selected as the final navigation directions:

$$\mathbf{d}_N^{\mathrm{view}} = \arg\min_m \frac{\mathbf{d}_N^{\mathrm{ATE}} \cdot \mathbf{d}_{Nm}^{\mathrm{view}}}{\|\mathbf{d}_N^{\mathrm{ATE}}\|\|\mathbf{d}_{Nm}^{\mathrm{view}}\|}, \tag{6}$$

and the final navigation point $\mathbf{x}_N^{\mathrm{view}}$ is calculated by

$$\mathbf{x}_N^{\mathrm{view}} = \mathbf{x}_N^{\mathrm{cluster}} + \eta \cdot \mathbf{d}_N^{\mathrm{view}}, \tag{7}$$

where $\eta$ is a hyper parameter representing the distance between the agent and the crowd cluster. NLBN ensures close-range, precise target observation and safe navigation, even in complex and unfamiliar environments. The potential navigation vectors, normal lines, and ATE view vectors are illustrated in Figure 2 (b).

### 3.3.4   Fine Detection and Counting (FDC)

Using the navigation points generated by NBLN, the agent travels from one point to another through path planning algorithms. Close-range and high-resolution RGB observations can be conducted upon

Table 2: Comparison with ZSON methods. ZECC achieves a trade-off between MAPE and TD.

| Method | MAPE (%) | TD (m) |
|---|---|---|
| FBE [45] + GL [47] | 57.19 ± 1.83 | 2513.06 ± 247.35 |
| FBE [45] + GD [39] | 53.38 ± 1.26 | 2513.06 ± 247.35 |
| CoW [9] + GL [47] | 52.75 ± 1.52 | 3449.51 ± 127.2 |
| CoW [9] + GD [39] | 46.01 ± 0.96 | 3449.51 ± 127.2 |
| OpenFMNav [24] + GL [47] | 60.57 ± 2.43 | 5069.64 ± 183.23 |
| OpenFMNav [24] + GD [39] | 49.41 ± 2.35 | 5069.64 ± 183.23 |
| ZECC | 18.71 ± 1.41 | 3722.45 ± 73.78 |

Table 3: Comparison with MVC methods. ZECC achieves a trade-off between MAPE and cost.

| Method | MAPE (%) | # of Cameras |
|---|---|---|
| MVF-10 [56] | 15.13 ± 0.00 | 1735.32 ± 0.00 |
| MVF-20 [56] | 39.92 ± 0.00 | 747.32 ± 0.00 |
| MVF-30 [56] | 61.43 ± 0.00 | 333.32 ± 0.00 |
| CountFormer-10 [34] | 12.8 ± 0.00 | 1735.32 ± 0.00 |
| CountFormer-20 [34] | 35.26 ± 0.00 | 747.32 ± 0.00 |
| CountFormer-30 [34] | 56.76 ± 0.00 | 333.32 ± 0.00 |
| ZECC | 18.71 ± 1.41 | 5 ± 0.00 |

reaching each navigation point. These observations are then utilized for detection models to perform precise target detection. The detection results are projected onto the global point cloud using the depth sensor and the agent's pose. To prevent repetitive detections, only one target is retained for each region within a specific scale. Ultimately, the result is calculated by counting the number of filtered targets.

# 4   Experiments

**Baselines.** We compare ZECC with exploration methods: Frontier-based exploration (FBE) [45], ZSON methods: CoW [9] and OpenFMNav [24], and multi-view counting (MVC) methods [56, 34].

**Metrics.** Mean Absolute Percentage Error (MAPE) is used to evaluate the counting performance: $\text{MAPE} = \frac{1}{M} \sum_{i=1}^{M} \left| \frac{y_i - \hat{y}_i}{y_i} \right| \times 100\%$ where $M$ is the quantity of testing environments, $\hat{y}_i$ and $y_i$ are the estimated count and the ground truth count, respectively.

For ZSON methods, the sum of Euclidean distance between adjacent navigation points along the agent traveling path is used to evaluate the travel distance (TD), which is defined as: $\text{TD} = \sum_{i=1}^{n-1} \|\mathbf{x}_{i+1} - \mathbf{x}_i\|$, where $n$ is the quantity of navigation points in one episode, $\mathbf{x}_i$ and $\mathbf{x}_{i+1}$ are the coordinates of the two adjacent navigation points, respectively. For the multi-view crowd counting method, we report the number of cameras required to achieve comparable performance to the proposed approach.

**Implementation Details.** MLLM used in ATE is GPT-4V [53]. [43] is used as path planning method. Altitude is 80m for HAE and 10m for LAE. The crowd density estimator in ATE is Generalized Loss (GL) [47], and the detection model in FDC is Grounding DINO (GD) [39]. For hyper parameters, navigation vector deg $\zeta$ is 15 °, density threshold $\kappa$ is 0.7, navigation point range $\eta$ is 8 m, and cluster size $\epsilon$ is 40. For the ZSON methods, the exploration step limitation is removed. When the methods finish an exploration step, they additionally get RGB-D captures of the environment. As exploration stops, they are equipped with GD or GL to conduct crowd counting using the projection method in FDC upon their captured images. For MVC, the scenes are divided into grids with diverse intervals, which are 10m, 20m, and 30m. Four cameras at poses of 0°, 90°, 180°, and 270° are placed on each grid intersection to obtain RGB captures. Then the captured images are sent to the MVC to get counting results. All methods are proceeded on a platform with IntelCorei9-14900KF, 128GBRAM, and NVIDIA GeForce RTX 4090 GPU.

# 5   Results

## 5.1   Overall Performance

**Comparison with ZSON methods.** We report performance in Table 2. ZECC offers the optimal balance between counting performance and cost. In contrast, FBE delivers the best TD but cannot perform a target detection function, which limits its capacity for fine observation. ZSON methods come with perception modules; however, they do not select a 3D fine observation point as NLBN, which means they cannot fully observe each individual in the crowd.

Agents utilizing counting models for crowd counting tend to perform slightly lower than agents using detection models. This is due to the increased sparsity of crowd observations as the agents get closer to the crowd. Crowd counting models are mainly trained on images depicting dense crowds, which results in limited generalization capability for scenarios with sparser crowds.

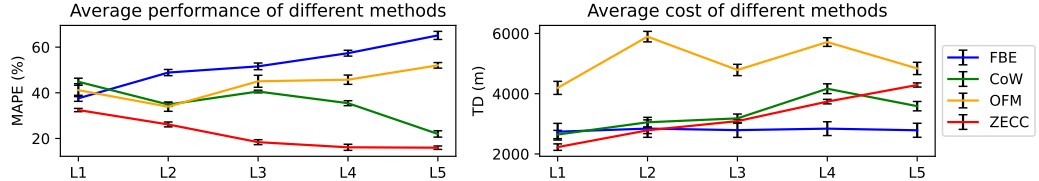

Figure 4: Performance and cost of ZECC and the baselines under different crowd density levels. L1-L5 refers to increasing density level. The figure demonstrates that ZECC achieves a balance between performance and exploration cost.

Table 4: The ablation study for ATE. ZECC achieves a better trade-off between performance and exploration efficiency (reducing 17% cost with 8% performance decline).

| Method | MAPE (%) | TD (m) |
|--------|----------|--------|
| w/o HAE | 17.46 | 4633.67 |
| w/o LAE | 88.08 | 1738.84 |
| ZECC | 18.91 | 3804.63 |

Table 5: The results of the ablation study for components in NLBN. ZECC achieves the best performance and success rate.

| Method | MAPE (%) | Successful rate (%) |
|--------|----------|---------------------|
| w/o NLBN | 65.19 | 100.00 |
| w/o NL | 98.44 | 8.33 |
| w/o VPS | 92.55 | 100.00 |
| w/o ATE-VPS | 99.49 | 1.45 |
| ZECC | 18.91 | 100.00 |

We analyze various environments with different crowd density levels and visualize the average performance and cost of the methods using the GD detector in these scenarios. The results are presented in Figure 4. Among the methods evaluated, ZECC demonstrates the best average performance across different density levels. As crowd density increases, the MLLM in ATE with NLBN becomes more effective at identifying and observing high-density crowds, resulting in improved performance. In contrast, other methods struggle to detect high-density crowds.

In terms of cost, ZECC's navigation points are influenced by crowd distribution and density. As the density level increases, the cost also rises. Although ZECC falls short of achieving the lowest cost in the last two density levels, its costs are still comparable to the baseline while ensuring effective counting performance. In contrast, other methods do not actively adjust navigation points. Their costs remain relatively stable, yet their performance is limited.

**Comparison with MVC.** The comparison with MVC is shown in Table 3. "-10" refers to MVC methoods using a grid interval of 10m. CountFormer-10 achieves the best performance, while ZECC still provides a close performance by reducing the camera used significantly. This illustrates the advantage of the active method over the multi-view method.

## 5.2 Ablation Study

**ATE.** We conducted ablation studies by removing specific components from ATE. The results are presented in Table 4. w/o HTE refers to using FBE + NLBN results for crowd counting. w/o LAE refers to fix the agent's altitude to HAE. w/o HAE performs best by exploring environments greedily, but results in a higher TD. ZECC achieves a better balance by conducting both HAE and LAE simultaneously. To further illustrate this trade-off, we conducted an experiment by fixing TD for FBE in w/o ATE and ATE in ZECC. Once the agent reaches a TD threshold, it will conduct NLBN using partial estimated crowd distribution. The result is shown in Figure 5 (a). ZECC demonstrates better performance with less cost when TD is limited in most cases, illustrating that ZECC can efficiently estimates global crowd distribution. On the other hand, w/o ATE is not effective when TD is limited.

**NLBN.** We then conducted ablation studies on NLBN and the results are presented in Table 5. w/o NLBN refers to using ATE results for crowd counting. w/o NL refers to not using normal line (NL) to calculate navigation points but use the cluster centers as navigation points. w/o VPS refers to not using view point selection (VPS) but select a point along the normal vector with $\eta$. w/o ATE-VPS refers to not using the ATE-view-vector-based view point selection (ATE-VPS), but randomly select a navigation vector from the potential navigation vectors. Successful rate indicates the ratio of the reachable navigation point reported by path planning algorithm. The findings indicate that omitting any component of NLBN results in a significant drop in either performance or the success rate. Without NLBN, the absence of optimized viewpoints causes ZECC to revert to a ZSON method. When both NL and FBE-VPS are removed, most navigation points end up being located on obstacles. Additionally, without VPS, the navigation points are positioned directly above the targets, causing the targets to fall out of the field of view.

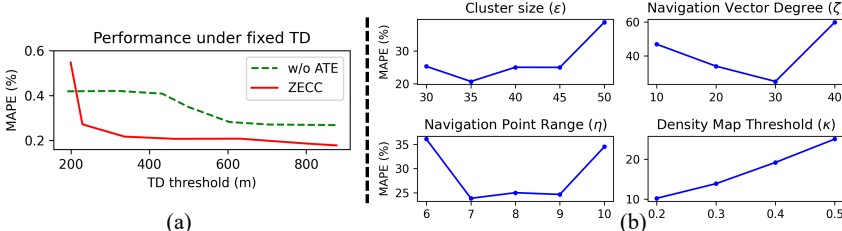

Figure 5: (a) Comparison of performance-cost trade-off. ZECC achieves a better trade-off when TD is limited. (b) The effect of four hyper parameters in ZECC. It shows that ZECC is effecive when the hyper-parameters are set in reasonable scopes.

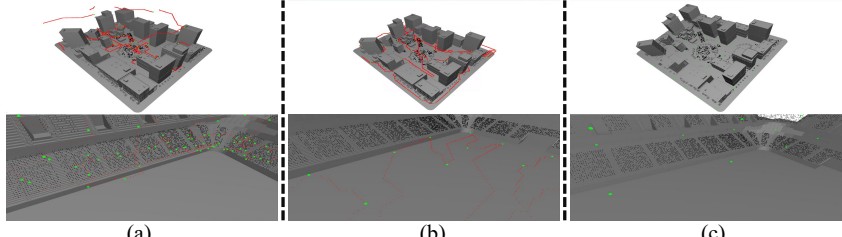

Figure 6: Navigation point (green) and trajectory (red) of different methods. Blcak dots are ground-truth. (a) ZECC. (b) OpenFMNAV. (c) MVC. ZECC shows less exploration in non-crowded areas while setting navigation points actively based on crowd distribution. Zoom in for a better view.

### 5.3 Hyper parameter Study

The influence of hyper parameters is studied, which include cluster size, navigation vector degree, navigation point range, and density map threshold. A gym-like scene featuring a densely packed crowd is utilized to test these parameters. The results are illustrated in Figure 5 (b).

**Cluster size.** Larger $\epsilon$ generates fewer observation with coarser detection, while low $\epsilon$ generates more navigation points and reduces efficiency (TD is 5125.44m for $\epsilon = 30$, 4065.02m for $\epsilon = 40$).

**Navigation vector degree.** The agent has a limited field of view for large $\zeta$ and is obstructed for low $\zeta$. This highlights the importance of the NLBN since the method generates robust navigation points for different scenes.

**Navigation point range.** At close range, the agent's field of view is restricted, and at long range, the agent is unable to gather detailed observations, which negatively impacts the subsequent target detection phase.

**Density map threshold.** A lower density map threshold leads to an expansion of the target area, resulting in an increased number of navigation points. However, this also raises the associated costs. For instance, when $\kappa = 0.5$, TD is 6408.02 m, increasing to 7272.02 m when $\kappa = 0.4$. This is a significant degradation in efficiency, yet the improvement in performance is not significant.

Generally, the influence of hyper parameter is consist with tuition. ZECC provides effective performance if they are not set to extreme value.

### 5.4 Case Study

We qualitatively illustrate how ZECC can alleviate occlusion and overlap while improving efficiency by comparing trajectory in two scenes with OpenFMNav and MVC-30. The results are shown in Figure 6. It shows that ZECC can effectively explore a complex occlusion environment, reduce exploration in low crowd density areas, and set navigation points based on crowd distribution. These features result in a trade-off between performance and cost.

## 6 Conclusion

In this study, we propose a task that enables interactive crowd counting: Embodied Crowd Counting (ECC). A simulator, the Embodied Crowd Counting Dataset (ECCD), is developed to enable related research for ECC. This dataset includes 60 diverse virtual environments with crowd density modeled

by a prior probability distribution, approximating reality. A method, Zero-shot Embodied Counting (ZECC), is proposed to verify this task. This is an active agent that can explore unknown environments without additional assistance. Active Top-down Exploration (ATE) is proposed to utilize Z-axis moving ability for exploration planning. This module is equipped with MLLM to enable active high altitude exploration (HAE) or low altitude exploration (LAE), balancing crowd counting performance and exploration cost. Normal-line based Navigation (NLBN) is proposed to select an optimized navigation point for crowd observation. This module generates a navigation point from the top-down view and maintains an angle, alleviating the overlap of the crowd. Simultaneously, the estimated navigation points enable obstacle avoidance and ensure that the crowd is in FOV. Experiment results show that ZECC achieves a balance between performance and cost compared to recent navigation agents. As the first work to propose ECC, we leave expending ECCD and ZECC to dynamic targets and real world application to future work, which are not considered by existing methods.

## 7 Acknowledgment

This work was supported by the Science and Technology Major Project of Jiangsu Province (No.BG2024041). This work was supported by the National Natural Science Foundation of China under Project 62406090. The work described in this paper was conducted in part by Dr ZHU Xinting, Jockey Club Global STEM Post-doctoral Fellow supported by The Hong Kong Jockey Club Charities Trust.

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

# Appendix

## A Broader Impacts

Further research and careful consideration are necessary when utilizing this technology, as the presented proposed method relies on the simulator, which may possess biases and could potentially result in negative societal impacts.

## B Visualization of ECCD

Visualization of several scenes in ECCD is shown in Figure 7 and Figure 8. ECCD offers a diverse range of scenarios, encompassing both large-scale outdoor settings and indoor environments. It provides a hierarchical 3D structure, with crowds distributed across various positions within the 3D space, posing challenges for algorithms.

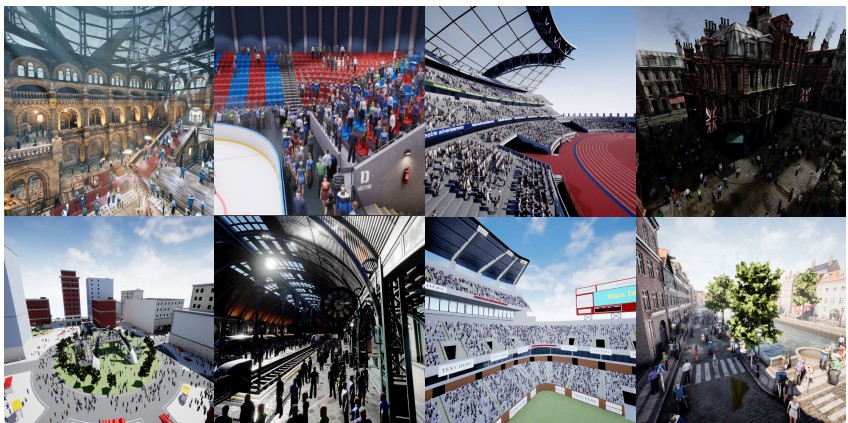

Figure 7: Visualization of environments sampled from ECCD.

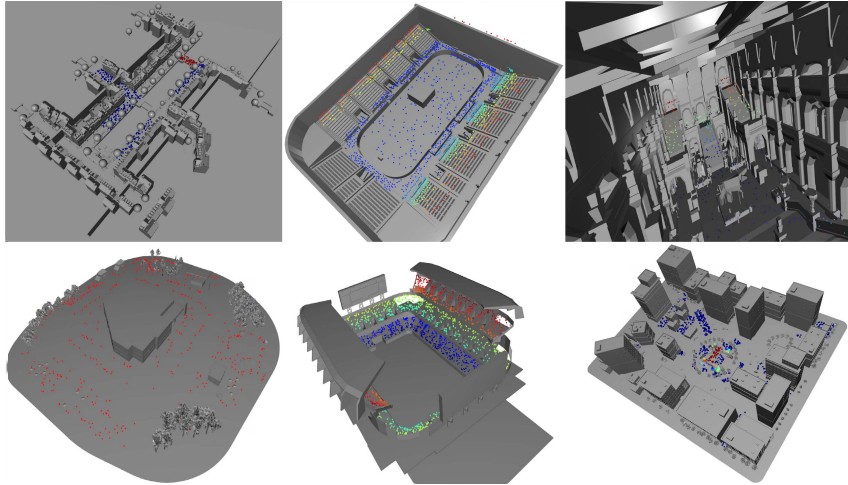

Figure 8: 3D structure of samples from ECCD. Colorful dots represent crowd ground truth. Different colors represent different heights.

# C Details of ZECC

## C.1 ATE

During HAE, the ATE receives RGB images captured from the bottom and surrounding areas of the agent. These images are organized into a panoramic view and transmitted to an MLLM in the form of a prompt. The image prompt and the text prompt are shown in Figure 9. The prompt utilizes the Chain of Thought mechanism to improve MLLM perception ability. Once the agent decides to switch to LAE, it will select a spare space directly under its horizontal location and conduct a landing operation. The pseudo-code is shown in Algorithm 1.

---

**Algorithm 1** Pseudo-Code of ATE

---

**Require:** Global distribution $D$, HAE map $M_H$, LAE map $M_L$, Prompt $I$
**Ensure:**
1: $t \leftarrow 0$;
2: $done1 \leftarrow False$;
3: $D \leftarrow None$
4: $M_H \leftarrow None$
5: $M_L \leftarrow None$
6: **while** not $done1$ **do**
7:     $O_t, p_t \leftarrow$ getState();
8:     $s_t \leftarrow$ MLLM $(O_t; I)$;
9:     $M_H \leftarrow$ updateMap $(O_t; p_t)$;
10:    **if** $s_t > 0.5$ **then**
11:        toLAE();
12:        $done2 \leftarrow False$;
13:        **while** not $done2$ **do**
14:            $O_f, p_f \leftarrow$ getState();
15:            $d_f \leftarrow$ P$(G(O_f), p_f)$;
16:            $M_L \leftarrow$ updateMap $(O_f; p_f)$;
17:            $D \leftarrow$ updateDistribution$(D, d_f)$;
18:            $done2 \leftarrow$ toUnexplored$(M_L)$;
19:        **end while**
20:        toHAE();
21:    **end if**
22:    $done1 \leftarrow$ toUnexplored$(M_H)$;
23:    $t \leftarrow t + 1$;
24: **end while**
25: **return** $D$

---

## C.2 NLBN

The method used to estimate the normal line of the crowd cluster plane is the Open3D package. The global crowd distribution is first divided into subareas using GMM cluster. Then, on each cluster, Open3D is used to estimate the normal line for each cluster. The normal lines are aligned with the vector (0,0,1). Then, the normal line at the cluster center is selected as the normal line of the cluster. Based on this representation, the navigation points can be obtained. The pseudo-code is shown in Algorithm 2.

## C.3 FDC

During FDC, the center of each detection box can be projected to the global crowd distribution by using the depth information. To re-identify targets, the space is divided into 3D voxels with size of 0.25 m. All the target with in a voxel is regared as the same target. This configuration is set for ZECC and all comparison methods.

**Algorithm 2** Pseudo-Code of NLBN

**Require:** Cluster size $\epsilon$, Navigation vector degree $\zeta$, Navigation point range $\eta$, Density map threshold $\kappa$, Global distribution $D$

**Ensure:**
1: $D \leftarrow \text{ATE}(D)$;
2: $\{\mathbf{x}_i^{\text{cluster}}, ..., \mathbf{x}_N^{\text{cluster}}\} \leftarrow \text{Cluster}(D)$;
3: $done \leftarrow False$;
4: **for** $i = 1$ **to** $N$ **do**
5: $\quad$ $\mathbf{d}_i^{\text{cluster}} \leftarrow \text{FitPlane}(\mathbf{x}_i^{\text{cluster}})$;
6: $\quad$ $\mathbf{x}_i^{\text{ATE}} \leftarrow \text{getViewpoint}(D)$;
7: $\quad$ Compute $\{\mathbf{d}_{i1}^{\text{view}}, ..., \mathbf{d}_{im}^{\text{view}}\}_i$ using Eq. (4);
8: $\quad$ Compute $\mathbf{d}_i^{\text{ATE}}$ using Eq. (5);
9: $\quad$ Compute $\mathbf{d}_i^{\text{view}}$ using Eq. (6);
10: $\quad$ Compute $\mathbf{x}_i^{\text{view}}$ using Eq. (7);
11: **end for**
12: **while** not $done$ **do**
13: $\quad$ $done \leftarrow \text{toNextNaviPoint}(\{\mathbf{x}_i^{\text{view}}, ..., \mathbf{x}_N^{\text{view}}\})$
14: **end while**

Figure 9: Prompt template used in ATE.

## D    Occlusion analysis on ZECC

In this section, we further test the ability of ZECC to alleviate occlusion in the complex environments. For each ground truth person, we first select the navigation point which has the minimal Euclidean distance to it, and then test whether occlusion occurs between the navigation point and the person. This is implemented by first projecting the global point cloud to the vector from navigation point to the person. Then, if the minimal distance between the global point cloud and the projected points is lower than a threshold, the person is obstructed. To formulate this, the unit vector from a person to its nearest navigation point is calculated as:

$$\mathbf{v}_i^{\mathrm{navi}} = \frac{\mathbf{P}_i^{\mathrm{navi}} - \mathbf{P}_i^{\mathrm{crowd}}}{\left\| \mathbf{P}_i^{\mathrm{navi}} - \mathbf{P}_i^{\mathrm{crowd}} \right\|}, \tag{8}$$

where $\mathbf{P}_i^{\mathrm{navi}}$ is coordinate of the navigation point and $\mathbf{P}_i^{\mathrm{crowd}}$ is the coordinate of the person. The vector from a global point cloud to the person is:

$$\mathbf{v}_{ij}^{\mathrm{global}} = \mathbf{P}_i^{\mathrm{navi}} - \mathbf{P}_j^{\mathrm{global}}, \tag{9}$$

where $\mathbf{P}_j^{\mathrm{global}}$ is a random coordinate from the global point cloud. The projected length is:

$$l_{ij} = \mathbf{v}_{ij}^{\mathrm{global}} \cdot \mathbf{v}_i^{\mathrm{navi}}, \tag{10}$$

and the projected point is:

$$\mathbf{P}_j^{\mathrm{proj}} = \mathbf{P}_i^{\mathrm{crowd}} + l_{ij} * \mathbf{v}_i^{\mathrm{navi}} \text{ s.t. } l_{ij} \geq 0 \wedge l_{ij} \leq \left\| \mathbf{v}_{ij}^{\mathrm{global}} \right\|. \tag{11}$$

The condition of determining the person is obstructed is:

$$\min_j \left\| \mathbf{P}_j^{proj} - \mathbf{P}_j^{global} \right\| \leq \lambda. \tag{12}$$

In the experiment, $\lambda$ is set to 0.5 m. We compare ZECC with OpenFMNav and MVC-30. The ratio between the number of obstructed person and the number of ground truth person of the three methods are shown in Table 6. It shows that ZECC suffers from the less occlusion, which benefits the crowd counting results.

## E    Failure case study

ZECC fails when MLLM makes wrong decision. This is mainly due to occlusion by buildings. Figure 10 shows an example of MLLM planning failure, where the agent does not choose to conduct LAE at a crowd area.

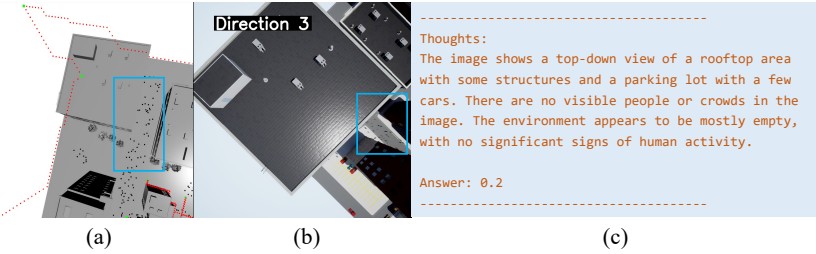

Figure 10: A case of ZECC failure. (a) Agent trajectory. (b) Top-down RGB. (c) MLLM reasoning. Blue box: ground truth crowd. Because of occlusion, MLLM failed to perceive the crowd during HAE.

## F    Quantitative results in real environment

To test ZECC in the real environment, we use two outdoor crowd scene to verify the proposed method. The drone model used is the Taobotics Q300. The captured images are transmitted back to a ground server using a remote communication protocol and fed into VGGT [49] to estimate relative point clouds and poses. The proposed NLBN is then employed to calculate relative navigation points. The

Table 6: Obstructed person ratio.

| Method | Ratio (%) |
|---|---|
| OpenFMNav | 42.81 |
| MVC-30 | 21.77 |
| ZECC | 4.52 |

Table 7: Performance of real scenarios.

| Methods | MAPE (%) Scenario 1 | Scenario 2 |
|---|---|---|
| w/o NLBN + GL | 41.41 | 57.76 |
| w/o NLBN + GD | 73.74 | 78.44 |
| NLBN + GL | 22.50 | 29.31 |
| NLBN + GD | 15.17 | 19.83 |

drone's absolute geographic coordinates and poses are recorded in real-time by its GPS. By leveraging the relationships between absolute and relative geographic coordinates and poses, the navigation points calculated by NLBN are mapped to absolute navigation points. After the ground server controls the drone to fly to these absolute navigation points, it captures images of the crowd. We utilize Grounding DINO and the crowd counting model Generalized Loss to perform crowd detection or counting on images captured from both distant and close-up perspectives, respectively. The ground truth for crowd annotations is manually labeled. The visualization of the drone observation before and after adjusting its position to the NLBN navigation point is shown in Figure 11. The quantitative results are shown in Table 7. Before applying NLBN, the drone can not obtain a clear view of the crowds in the target area. The GL estimation result is fuzzy, and due to crowd overlap, people in the back rows are obstructed by those in the front rows. This results in the corrupted detection of GD. By adjusting to the NLBN navigation point, the drone is able to observe the target area from a top-down perspective, alleviating occlusion of people in back rows and improving counting performance significantly. This demonstrates how NLBN addresses occlusion.

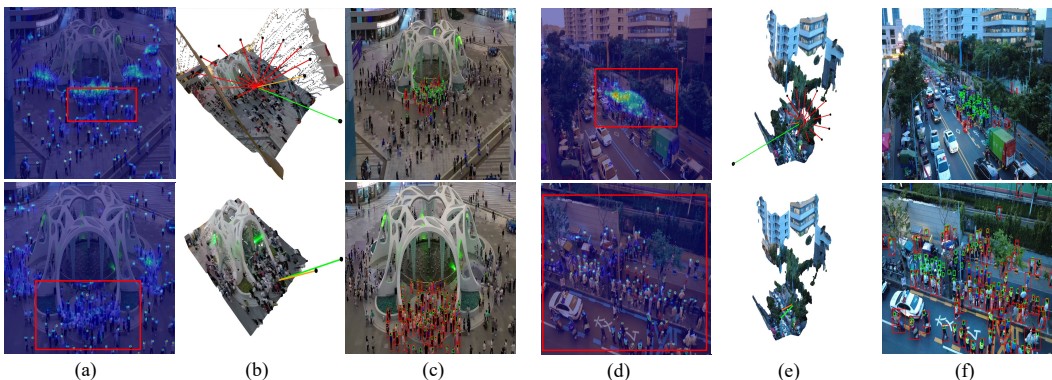

(a)          (b)          (c)          (d)          (e)          (f)

Figure 11: Quantitative results in real environment. (a), (d) Output of GL. The marked areas indicate the target zone. (b), (e) Point cloud of the scene. Red lines: candidate view vector. Green lines: the drone view vector. Orange lines: selected view vector. (c), (f) GD detection results. Red boxes: detected crowds. Green boxes: ground truth crowds. First row: before applying NLBN. Second row: after applying NLBN. (a), (b) and (c) are Scenario 1 and (d), (e) and (f) are Scenario 2. By adjusting the camera position to the NLBN navigation point, the target detection result is improved.

