# OpenReview forum: "Embodied Crowd Counting"
_NeurIPS.cc/2025/Conference — NeurIPS 2025 poster_

### Official Review · Reviewer_nYLZ · 2025-06-19

**Clarity:** 3
**Significance:** 4
**Originality:** 3
**Rating:** 6
**Confidence:** 5

**Summary:**

This paper proposes a new task called Embodied Crowd Counting (ECC). This task introduces an active camera setting into crowd counting, aiming to address the fundamental occlusion issue in traditional image-based crowd counting. To enable the task, this paper proposes Embodied Crowd Counting Dataset (ECCD), a simulation environment combines 60 large scale scenes and large number of crowds simultaneously. The crowd distribution is modeled by Poisson Point Process, a method to mimic realistic crowd distribution. Based on ECCD, an agentic method, Zero-shot Embodied Crowd Counting (ZECC), is proposed. This method first uses Active Top-down Exploration (ATE), a module based on Multi-modal Large Language Model (MLLM), to efficiently estimate coarse crowd distribution by adjusting exploration altitude. Then, Normal-line Based Navigation (NLBN) is used to conduct fine analysis of the crowd structure to avoid potential occlusion and enhance visibility, resulting in improved counting performance. Extensive experiments show that ZECC achieves strong performance against existing navigation-based methods and image-based counting methods while maintaining efficiency.

**Questions:**

See weaknesses.

**Ethical Concerns:**

["NO or VERY MINOR ethics concerns only"]

**Final Justification:**

Thanks for the rebuttal. My major concerns have been addressed well. I tend to accept this paper.

**Limitations:**

The limitations are described in the paper.

**Quality:**

4

**Strengths And Weaknesses:**

Strengths:
1)	The proposed ECC is novel. The introduction of embodied agent setting to crowd counting is well-motivated.
2)	ECCD is significantly different from existing dataset in both crowd counting and navigation with interactive ability and large target quantity and complex distribution.
3)	ZECC utilizes vertical movement to balance exploration cost and performance while analyzing crowd considering its 3D structure, which are not considered by existing methods and provides new perspectives in both navigation and crowd counting.
4)	The experiment results are comprehensive. The authors have demonstrated effectiveness of ZECC by comparing with existing ZSON and crowd counting methods, and the effectiveness of ATE and NLBN and the performance-cost trade-off are also illustrated. More experimental results in the Appendix further give analysis on the mechanism of ZECC reducing occlusion. These results comprehensively demonstrate the effectiveness of ZECC.
5)	Overall, this paper provides a novel perspective to both embodied navigation and crowd counting, and demonstrates its effectiveness convincedly, which can promote further progress in the fields.


Weaknesses:
1)	The paper does not explain ‘VPS’ and ‘ATE-VPS’ in Table 5, which makes this experiment confusing. This experiment seems to demonstrate the effectiveness of ATE viewpoint; however, a detailed explanation is necessary.
2)	ZECC contains numerous calculation steps and components such as ATE, NLBN, VPS, etc. It is suggested to include an algorithm chart for clarity.
3)	In the failure case study, the author mentioned that ATE is vulnerable in some occlusion cases. This may restrict performance of ZECC. A further discussion on this limitation would strength the paper.
4)	Minor weaknesses: The authors should define the task names in Table 1. ‘ZSON’ in line 132 is not clearly defined when first time used. Also, the reference is missing.
5)	In line 225, a space is missing between ‘Figure 2’ and ‘(b)’.

---

> ### Author Rebuttal · Authors · 2025-07-31
>
> > Q1: The paper does not explain ‘VPS’ and ‘ATE-VPS’ in Table 5, which makes this experiment confusing. This experiment seems to demonstrate the effectiveness of ATE viewpoint; however, a detailed explanation is necessary.
>
>  A1: We apologize for this typo. 'w/o VPS' refers to not using view point selection (VPS) but select a point along the normal
>  vector with **ε**. w/o ATE-VPS refers to not using the ATE-
> view-vector-based view point selection (ATE-VPS), but
> randomly select a navigation vector from the potential
> navigation vectors. We will add these explanation to the revised paper.
>
>  > Q2: ZECC contains numerous calculation steps and components such as ATE, NLBN, VPS, etc. It is suggested to include an algorithm chart for clarity.
>
>  A2: We provide the algorithm in Algorithm 1 and Algorithm 2. We will add these explanation to the revised paper.
>
> Algorithm 1 Pseudo-Code of ATE
>
> **Require:** Global distribution $D$, HAE map $M_H$, LAE map $M_L$, Prompt $I$
>
> **Ensure:**
>
> 1: $t \leftarrow 0$;
>
> 2: $done1 \leftarrow False$;
>
> 3: $D \leftarrow None$
>
> 4: $M_H \leftarrow None$
>
> 5: $M_L \leftarrow None$
>
> 6: **while** not $done1$ **do**
>
> 7:   $O_t, p_t \leftarrow getState()$;
>
> 8:   $s_t \leftarrow MLLM(O_t; I)$;
>
> 9:   $M_H \leftarrow updateMap(O_t; p_t)$;
>
> 10:   **if** $s_t > 0.5$ **then**
>
> 11:     $toLAE()$;
>
> 12:     $done2 \leftarrow False$;
>
> 13:     **while** not $done2$ **do**
>
> 14:       $O_f, p_f \leftarrow getState()$;
>
> 15:       $d_f \leftarrow P(G(O_f), p_f)$;
>
> 16:       $M_L \leftarrow updateMap(O_f; p_f)$;
>
> 17:       $D \leftarrow updateDistribution(D, d_f)$;
>
> 18:       $done2 \leftarrow toUnexplored(M_L)$;
>
> 19:     **end while**
>
> 20:   **end if**
>
> 21:   $done1 \leftarrow toUnexplored(M_H)$;
>
> 22:   $t \leftarrow t + 1$;
>
> 23: **end while**
>
> 24: **return** $D$
>
>
> Algorithm 2 Pseudo-Code of NLBN
>
> **Require:** Cluster size $\epsilon$, Navigation vector degree $\zeta$, Navigation point range $\eta$, Density map threshold $\kappa$, Global distribution $D$
>
> **Ensure:**
>
> 1: $D \leftarrow \text{ATE}(D)$;
>
> 2: $\{ \mathbf{x}_i^{\text{cluster}}, \ldots, \mathbf{x}_N^{\text{cluster}} \} \leftarrow \text{Cluster}(D)$;
>
> 3: $done \leftarrow False$;
>
> 4: **for** $i = 1$ to $N$ **do**
>
> 5:   $d_i^{\text{cluster}} \leftarrow \text{FitPlane}(\mathbf{x}_i^{\text{cluster}})$;
>
> 6:   $\mathbf{x}_i^{\text{ATE}} \leftarrow \text{getViewpoint}(D)$;
>
> 7:   Compute $\{ d_{i1}^{\text{view}}, \ldots, d_{im}^{\text{view}} \}_i$ using Eq. (4);
>
> 8:   Compute $d_i^{\text{ATE}}$ using Eq. (5);
>
> 9:   Compute $d_i^{\text{view}}$ using Eq. (6);
>
> 10:   Compute $\mathbf{x}_i^{\text{view}}$ using Eq. (7);
>
> 11: **end for**
>
> 12: **while** not $done$ **do**
>
> 13:   $done \leftarrow \text{toNextNaviPoint}(\{ \mathbf{x}_i^{\text{view}}, \ldots, \mathbf{x}_N^{\text{view}} \})$
>
> 14: **end while**
>
>  > Q3: In the failure case study, the author mentioned that ATE is vulnerable in some occlusion cases. This may restrict performance of ZECC. A further discussion on this limitation would strength the paper.
>
>  A3: We believe that this is mainly due to the fact that current ATE lacks the ability to identify positions that can effectively observe positions relevant to targets, such as roads or squares, from a top-down view, resulting in navigation to irrelevant areas, such as rooftops. We will study this challenge in our future work.
>
>  > Q4: Minor weaknesses: The authors should define the task names in Table 1. ‘ZSON’ in line 132 is not clearly defined when first time used. Also, the reference is missing.
>
>  A4: We apologize for this typo. ZSON is defined in [1]. We will add this definition to the revised paper.
>
>  > Q5: In line 225, a space is missing between ‘Figure 2’ and ‘(b)’.
>
>  A5: We apologize for this typo. We will revise this part in the paper.
>
>  [1] Majumdar, Arjun et al. “ZSON: Zero-Shot Object-Goal Navigation using Multimodal Goal Embeddings.” ArXiv abs/2206.12403 (2022): n. pag.

---

> > ### Comment · Reviewer_nYLZ · 2025-08-03
> >
> > Thanks for the rebuttal. My major concerns have been addressed well. I tend to accept this paper.

---

> > > ### Author Response · Authors · 2025-08-04
> > >
> > > We deeply appreciate the time and effort you invested in the evaluation of our paper. We are pleased to have addressed your concerns. We will add these results in the revision. Thank you again for your valuable suggestions and hope the response can enhance the quality and evaluation of the work.

---

### Official Review · Reviewer_dZWy · 2025-07-01

**Clarity:** 3
**Significance:** 2
**Originality:** 3
**Rating:** 4
**Confidence:** 4

**Summary:**

The paper introduces Embodied Crowd Counting (ECC), a new task that frames crowd counting as an active vision problem for a drone agent. The papers build ECCD, a 60-scene Unreal-Engine simulator that combines large outdoor areas with up to 15k people. They propose ZECC, a zero-shot baseline, including Active Top-down Exploration, Normal-Line-Based Navigation, and Fine Detections. Experiments show that the proposed method delivers a favourable trade-off between counting accuracy and navigation cost when compared with classical frontier exploration, zero-shot object-navigation agents, and a multi-view counting baseline.

**Questions:**

Please refer to weaknesses.

**Ethical Concerns:**

["NO or VERY MINOR ethics concerns only"]

**Final Justification:**

Although I think it is far away to real embodied application, the research is exploratory study. I revise my rating to positive.

**Limitations:**

No, there are no real-world experiments to implement this model.

**Quality:**

3

**Strengths And Weaknesses:**

Strengths
1. The paper defines a novel task—Embodied Crowd Counting—that recasts crowd analysis as an active-vision problem for mobile agents.
2. By bridging embodied AI and crowd counting, the work opens a new research direction and provides resources that can stimulate further studies in both communities.

Weaknesses
1. All evaluations are confined to the synthetic ECCD simulator. The real-world transferability of both the dataset and the ZECC policy remains unverified.
2. Please show the reasons for building embodied crowd counting instead of embodied small object detection. I think the latter is more valuable with location information.
3. The crowd generator relies on a simple Poisson point process, which ignores structured formations common in real gatherings (queues, flows, social spacing).
4. Comparative experiments omit recent multi-view crowd-counting methods (“Cross-View Cross-Scene Multi-View Crowd Counting,” and “CountFormer: Multi-View Crowd Counting Transformer”), making it hard to judge the improvement of ZECC.
5. ZSON appears without definition.

---

> ### Author Rebuttal · Authors · 2025-07-31
>
> > Q1: All evaluations are confined to the synthetic ECCD simulator.
>
>  A1: Thank you for your review to improve our work. We have provided a real-world application case in &zwnj;**Appendix 6**&zwnj; to test ZECC's capability in reducing occlusions in densely crowded scenarios. Besides, we provide results in another real-world scenario, which is denser compared to the existing one. The drone model used is the Taobotics Q300. The captured images are transmitted back to a ground server using a remote communication protocol and fed into VGGT [1] to estimate relative point clouds and poses. The proposed NLBN is then employed to calculate relative navigation points. The drone's absolute geographic coordinates and poses are recorded in real-time by its GPS. By leveraging the relationships between absolute and relative geographic coordinates and poses, the navigation points calculated by NLBN are mapped to absolute navigation points. After the ground server controls the drone to fly to these absolute navigation points, it captures images of the crowd. We utilize Grounding DINO and the crowd counting model Generalized Loss to perform crowd detection or counting on images captured from both distant and close-up perspectives, respectively. The ground truth for crowd annotations is manually labeled.
>
>  > Q2: Please show the reasons for building embodied crowd counting instead of embodied small object detection. I think the latter is more valuable with location information.
>
>  A2: It is indeed that ZECC leverages the concept of small object detection. However, we think that crowds often appear with heavy density and occlusion, which makes conducting embodied tasks on crowds more challenging. These complex distributions require additional techniques like NLBN to generate fine viewpoints to gain accurate counting results. We will broaden our scope to general small object detection in our future work.
>
>  > Q3: The crowd generator relies on a simple Poisson point process, which ignores structured formations common in real gatherings (queues, flows, social spacing).
>
>  A3: The current ECCD includes scenes that simulate real crowd distribution, such as queues and flows. As mentioned in &zwnj;**line 159**&zwnj; in the manuscript, the Poisson point process only determines the number and distribution of individuals &zwnj;**in a block**&zwnj;, and the shape and distribution of the blocks are adjusted by human experts according to the environment semantics. In this way, crowds in street scenes appear to be in queues, and crowds in intersections appear to be in flows.
>
>  > Q4: Comparative experiments omit recent multi-view crowd-counting methods ("Cross-View Cross-Scene Multi-View Crowd Counting," and "CountFormer: Multi-View Crowd Counting Transformer"), making it hard to judge the improvement of ZECC.
>
>  A4: Since no open source code for [2] is found, we compare with [3] using the same settings as MVC. The weight of the models are fixed during testing. The results are shown in Table 1. Although recent MVC methods show improvement, they still suffer from significant degradation when the number of cameras decreases. MVC methods cannot fundamentally solve the occlusion issue because of fixed cameras. This result illustrates the necessity of embodied crowd counting.
>
> Table 1: Comparison with different MVC methods
> | Method          | MAPE (%)    | \# of Cameras |
> |-----------------|-------------|----------------|
> | MVC-10          | 15.13 ± 0.00| 1735.32 ± 0.00 |
> | MVC-20          | 39.92 ± 0.00| 747.32 ± 0.00  |
> | MVC-30          | 61.43 ± 0.00| 333.32 ± 0.00  |
> | CountFormer-10  | 12.8 ± 0.00 | 1735.32 ± 0.00 |
> | CountFormer-20  | 35.26 ± 0.00| 747.32 ± 0.00  |
> | CountFormer-30  | 56.76 ± 0.00| 333.32 ± 0.00  |
> | ZECC            | 18.71 ± 1.41| 5 ± 0.00       |
>
> > Q5: ZSON appears without definition.
>
> A5: We apologize for this typo mistake. ZSON is defined in [4]. We will add this definition to the revised paper.
>
> [1] Wang, Jianyuan et al. "VGGT: Visual Geometry Grounded Transformer." Computer Vision and Pattern Recognition (2025).
>
> [2] Zhang, Qi et al. "Cross-View Cross-Scene Multi-View Crowd Counting." 2021 IEEE/CVF Conference on Computer Vision and Pattern Recognition (CVPR) (2021): 557-567.
>
> [3] Mo, Hong et al. "CountFormer: Multi-View Crowd Counting Transformer." ArXiv abs/2407.02047 (2024): n. pag.
>
> [4] Majumdar, Arjun et al. “ZSON: Zero-Shot Object-Goal Navigation using Multimodal Goal Embeddings.” ArXiv abs/2206.12403 (2022): n. pag.

---

> ### Author Response · Authors · 2025-08-05
>
> Dear Reviewer,
>
> Thank you again for your time and effort invested in the evaluation of our paper. We would like to ensure that we have addressed all your concerns satisfactorily. If there are any additional feedback you'd like us to consider, please let us know. Your insights are invaluable to us, and we're eager to address any remaining issues to improve our work.
>
> Thank you again for your time and effort in reviewing our paper.

---

> ### Comment · Reviewer_dZWy · 2025-08-06
>
> The authors have been resolved my concerns. Although I think it is far away to real embodied application, the research is exploratory study. I revise my rating to positive.

---

> > ### Author Response · Authors · 2025-08-06
> >
> > We deeply appreciate the time and effort you invested in the evaluation of our paper. We are pleased to have addressed your concerns. We will conduct further study on sim-to-real techniques tailored to the ECC task to enhance its real application and discuss this aspect in the revised paper. Thank you again for your valuable suggestions, which are crucial for improving the quality of the paper.

---

### Official Review · Reviewer_szRu · 2025-07-02

**Clarity:** 3
**Significance:** 3
**Originality:** 3
**Rating:** 4
**Confidence:** 5

**Summary:**

The paper introduces a task called Embodied Crowd Counting (ECC), which tackles the long-standing issue of occlusion in crowd counting by shifting from passive to active perception. Instead of relying on static camera views, the authors propose deploying a mobile aerial agent (like a drone) that can dynamically navigate large-scale outdoor environments.

To support this task, they create ECCD, a realistic simulator featuring 60 virtual city-scale scenes with up to 15,000 individuals per scene. It uses a Poisson Point Process to simulate authentic crowd distributions. The authors also propose ZECC, a zero-shot agent that explores scenes using a two-stage approach: a Multi-modal Large Language Model (MLLM)-guided Active Top-down Exploration (ATE) for coarse crowd estimation and a Normal-line Based Navigation (NLBN) for refining observation points.

Experiments claim to show ZECC achieves a strong balance between accuracy and efficiency compared to existing methods. Results show a commendable trade-off between performance and exploration cost, making a strong case for the real-world relevance of the proposed approach.

**Questions:**

1. Statistical Significance and Variance Reporting

Could the authors include statistical metrics (e.g., standard deviation, confidence intervals) for the main experimental results, particularly across density levels or repeated trials with different random seeds or initial positions? Without this, it’s hard to evaluate the robustness of ZECC, especially since it leverages MLLMs, which can behave non-deterministically.

2. Explainability and Failure Cases

Could the authors provide visualizations or examples of failure modes (e.g., when ZECC undercounts/overcounts or mis-navigates due to occlusion or poor density estimation)? Including 2 to 3 such examples with interpretation (e.g., where ATE failed to prioritize a dense area, or NLBN chose suboptimal angles).

3. The use of GPT-4V for determining exploration altitude is clever. But how consistent are its decisions? Is it queried once per scene or at every step?

**Ethical Concerns:**

["NO or VERY MINOR ethics concerns only"]

**Final Justification:**

I am maintaining my original score, which was already at the highest end in regards to the quality of the paper/work. The authors have provided clear and satisfactory responses during the rebuttal phase. While the questions raised were important, they served more as confirmation and follow-up to ensure the scientific rigor of the work, rather than indicating any serious concerns. The authors' replies demonstrated a strong understanding of the issues and confirmed that they are approaching the problem in a sound and principled manner. As such, I see no unresolved issues, and I continue to support acceptance of this submission.

**Limitations:**

yes

**Paper Formatting Concerns:**

The paper is aligned with the formatting guidelines.

**Quality:**

3

**Strengths And Weaknesses:**

Quality:

The technical execution is thorough. ECCD is thoughtfully designed to simulate complex crowd distributions across diverse, large-scale scenes, addressing key limitations in prior datasets. The experimental section is well-detailed, with comparisons against standard baselines (ZSON, MVC), and ablations of ATE and NLBN components. However, the lack of statistical significance reporting (e.g., error bars, variance) is a notable gap, especially given the dynamic nature of the exploration process and the use of non-deterministic MLLMs.


Clarity:

The paper is generally well written, with clear illustrations. The architectural design of ZECC is easy to follow, and the motivation behind each module is well explained. That said, some parts, particularly the mathematical formulations for NLBN could benefit from more intuitive grounding or simplified descriptions.


Significance:

This work has high potential impact. By bridging the gap between embodied navigation and crowd analysis, it opens up a new research direction that aligns with real-world deployment scenarios. ECCD could serve as a benchmark to catalyze further innovation in this domain. That said, since the entire framework is still simulated, questions around transferability to real-world deployment (e.g., noisy sensors, unpredictable crowds) remain unanswered.


Originality:

The task definition, dataset design, and the approach are all fresh and well-motivated. While elements like Z-axis planning or crowd modeling with Poisson processes are known, their integration into a unified embodied framework for crowd counting is novel and impactful. ZECC’s zero-shot formulation is particularly a contribution, leveraging MLLMs without heavy retraining.

---

> ### Author Rebuttal · Authors · 2025-07-31
>
> > Q1: Could the authors include statistical metrics.
>
>  A1: Thank you for the reviews to improve our work. We have updated the main experimental results with standard deviation in Table 1, Table 2, and Table 3. In each experiment, we generate three random starting positions within the navigable area. Additionally, the random seed is changed three times for each starting position, resulting in a total of nine trials for every experiment. The results show that the performance of ZECC is consistent.
>
>  Table 1: Statistical performances.
>  | Method                                                       | MAPE (%)     | TD (m)           |
> |--------------------------------------------------------------|--------------|------------------|
> | FBE + GL     | 57.19 ± 1.83 | 2513.06 ± 247.35 |
> | FBE + GD      | 53.38 ± 1.26 | 2513.06 ± 247.35 |
> | CoW + GL | 52.75 ± 1.52 | 3449.51 ± 127.2  |
> | CoW + GD | 46.01 ± 0.96 | 3449.51 ± 127.2  |
> | OpenFMNav + GL| 60.57 ± 2.43 | 5069.64 ± 183.23 |
> | OpenFMNav + GD | 49.41 ± 2.35 | 5069.64 ± 183.23 |
> | ZECC                                                         | 18.71 ± 1.41 | 3722.45 ± 73.78  |
>
>
> Table 2: MAPE under different crowd density
>
> | Methods   | L5            | L4            | L3            | L2            | L1            |
> |-----------|---------------|---------------|---------------|---------------|---------------|
> | FBE       | 65.13 ± 1.77  | 57.35 ± 1.28  | 51.58 ± 1.46  | 48.86 ± 1.36  | 37.58 ± 1.32  |
> | CoW       | 21.98 ± 1.39  | 35.45 ± 1.09  | 40.56 ± 0.57  | 34.85 ± 0.79  | 44.84 ± 1.55  |
> | OpenFMNav | 52.02 ± 1.22  | 45.73 ± 2.03  | 45.04 ± 2.62  | 33.93 ± 2.04  | 41.14 ± 2.83  |
> | ZECC      | 15.96 ± 0.75  | 16.13 ± 1.34  | 18.38 ± 1.09  | 26.14 ± 1.07  | 32.45 ± 0.71  |
>
> Table 3: TD under different crowd density levels
>
> | Methods   | L5            | L4            | L3            | L2            | L1            |
> |-----------|---------------|---------------|---------------|---------------|---------------|
> | FBE       | 2786.23 ± 233.01 | 2840.13 ± 229.27 | 2791.12 ± 240.75 | 2841.87 ± 289.77 | 2738.76 ± 276.85 |
> | CoW       | 3586.67 ± 155.39 | 4165.59 ± 162.44 | 3181.45 ± 143.66 | 3051.42 ± 164.31 | 2645.32 ± 137.58 |
> | OpenFMNav | 4838.67 ± 203.53 | 5714.58 ± 144.36 | 4787.69 ± 187.65 | 5894.42 ± 174.61 | 4193.41 ± 216.72 |
> | ZECC      | 4289.07 ± 68.94  | 3737.72 ± 75.73  | 3090.17 ± 53.6   | 2785.6 ± 109.92  | 2227.3 ± 107.59  |
>
>
>  > Q2: Could the authors provide visualizations or examples of failure modes.
>
>  A2: We provide a failure case in &zwnj;**Section 5 in the Appendix**&zwnj;. It shows that a rooftop occludes the MLLM and cannot realize the crowd beside the building. This is mainly due to the fact that current ATE lacks the ability to identify positions that can effectively observe positions relevant to targets, such as roads or squares, from a top-down view, resulting in navigation to irrelevant areas, such as rooftops. We will study this challenge in our future work.
>
>  NLBN sometimes chooses suboptimal positions. In our experiments, this is mainly caused by the wrong detection results of the crowd counting model used to estimate the crowd distribution. It incorrectly identifies trees as a crowd in some cases, making NLBN generate navigation points near these target irrelevant areas. This can be solved by introducing a better target detection model to filter these cases or training a more robust crowd distribution estimator.
>
>  > Q3: But how consistent are its decisions? Is it queried once per scene or at every step?
>
>  A3: We provide statistical metrics of ZECC, and from the results in Table 2, Table 3 and Table 4, it shows that the result is consistent. We believe this is because crowds can be easily identified during HAE in most cases, as the drone's top-down view provides a clear sight, enabling the MLLM to make consistent decisions.
>
>  The MLLM used in ZECC is used to decide whether to conduct LAE in each HAE step.

---

> > ### Comment · Reviewer_szRu · 2025-08-01
> > **Questions and doubts are well-clarified**
> >
> > The authors have made clear all of the questions asked, thank you for that. There is no further questions from my end as a reviewer.

---

> > > ### Author Response · Authors · 2025-08-04
> > >
> > > We deeply appreciate the time and effort you invested in the evaluation of our paper. We are pleased to have addressed your concerns. We will add these results in the revision. Thank you again for your valuable suggestions and hope the response can enhance the quality and evaluation of the work.

---

### Official Review · Reviewer_RciT · 2025-07-03

**Clarity:** 3
**Significance:** 2
**Originality:** 2
**Rating:** 4
**Confidence:** 3

**Summary:**

In this paper, the authors propose a novel method for crowd counting using embodied agents that actively navigate environments to overcome occlusion and limited visibility—key challenges for traditional single-frame approaches. They introduce ZECC, a zero-shot agent guided by a vision-language model for effective exploration, and a geometry-based strategy for selecting optimal viewpoints for counting. For the novel Embodied Crowd Counting task, they build ECCD, a large-scale interactive dataset with realistic outdoor and variable crowd scenes. Benchmarking against other baselines, ZECC achieves better accuracy, demonstrating the effectiveness of active, embodied exploration for crowd analysis.

**Questions:**

* Can the authors provide insights or preliminary evidence on real-world applicability?  Given that the entire pipeline is evaluated in simulation, it remains unclear how well ZECC would transfer to real environments. Could the authors either discuss the expected challenges in real-world deployment or share preliminary qualitative results on real data? Can ECCD be enriched with real scenes, or made more realistic using existing sim-to-real approaches?

* How critical is the vision-language model (GPT-4V) to the success of ZECC?  GPT-4V plays a key role in altitude decision-making during exploration, but it is a large, closed model. Could the authors clarify how much performance depends on this specific choice, and whether smaller or open-source alternatives (e.g., BLIP-2, MiniGPT) could be used effectively?

**Ethical Concerns:**

["NO or VERY MINOR ethics concerns only"]

**Final Justification:**

I want to thank the authors for the provided rebuttal. The answers to my questions are clear, but I would like to maintain my original rating. The authors agreed that the method "mainly focuses on system-level innovation", which is an important contribution by itself (also considering the ECCD environment), and I reflect this in my positive original rating. However, methodologically, it undermines the novelty of the proposed pipeline, which is mostly a combination of existing algorithms and methods.
Regarding the real-world evaluations, as the authors noted, 3DGS-based solutions or simulators could have been considered, as they are among the popular simulation environments used in the industry for such (closed-loop) evaluations. This would have made the paper significantly more solid.

**Limitations:**

yes

**Quality:**

2

**Strengths And Weaknesses:**

Strengths:

* The paper introduces Embodied Crowd Counting (ECC), which frames crowd counting as an active exploration task using embodied agents. This formulation is more aligned with real-world scenarios (e.g., drones or mobile robots) than traditional static image-based approaches, which often fail under occlusion or poor viewpoints.

* The authors release ECCD, a large-scale synthetic dataset that includes diverse and realistic outdoor environments with varying crowd densities. The dataset enables reproducible benchmarking and is a valuable contribution for the research community exploring active perception and crowd analysis.

* The authors propose an effective method ZECC for the ECC task, which significantly outperforms other baselines across both accuracy and efficiency metrics.

Weaknesses:

* The paper's method and experiments are entirely conducted in a synthetic environment (ECCD), without any real-world evaluation. While the simulator is well-designed, transferring performance gains from simulation to real-world crowd scenarios is non-trivial due to domain gaps in appearance, noise, and agent behavior. This limits the practical significance and generalizability of the results, and applying the method to real data would likely require substantial adaptation.

* The method is primarily a composition of existing components—such as pretrained vision-language models for navigation, classical path planning (A*), and standard crowd detection and clustering techniques. While the integration is thoughtful and effective for the proposed task, the paper introduces limited methodological novelty in terms of new algorithms. The strength lies more in system-level design than in technical innovation at the model level.

---

> ### Author Rebuttal · Authors · 2025-07-31
>
> > Q1: The method and experiments are entirely conducted in a synthetic environment (ECCD), without any real-world evaluation.
>
>  A1: Thank you for your review to improve our work. We have provided a real-world application case in &zwnj;**Appendix 6**&zwnj; to test ZECC's capability in reducing occlusions in densely crowded scenarios. Besides, we provide results in another real-world scenario, which is denser compared to the existing one. The drone model used is the Taobotics Q300. The captured images are transmitted back to a ground server using a remote communication protocol and fed into VGGT [1] to estimate relative point clouds and poses. The proposed NLBN is then employed to calculate relative navigation points. The drone's absolute geographic coordinates and poses are recorded in real-time by its GPS. By leveraging the relationships between absolute and relative geographic coordinates and poses, the navigation points calculated by NLBN are mapped to absolute navigation points. After the ground server controls the drone to fly to these absolute navigation points, it captures images of the crowd. We utilize Grounding DINO and the crowd counting model Generalized Loss to perform crowd detection or counting on images captured from both distant and close-up perspectives, respectively. The ground truth for crowd annotations is manually labeled.
>
> The results in Table 1 demonstrate that the methods with NLBN perform significantly better than baselines, which illustrates the real-world generalization ability of our method. We will add these results to the revised paper.
>
> Table 1: Performance before and after applying NLBN in the real scenes.
>
> | Methods | MAPE (%) in Scenario 1 | MAPE (%) in Scenario 2 |
> | --- | --- | --- |
> | w/o NLBN + GL  | 41.41 | 57.76 |
> | w/o NLBN + GD  | 73.74 | 78.44 |
> | NLBN + GL  | 22.50 | 29.31 |
> | NLBN + GD  | 15.17 | 19.83 |
>
>
>  > Q2: The method is primarily a composition of existing components.
>
>  A2: We agree that this work mainly focuses on system-level innovation; however, it also provides novel algorithms to address the occlusion issue and a height adjustment method to improve exploration efficiency. Also, we propose a new task, ECC, and a new platform, ECCD, that can inspire future research in this area.
>
> &zwnj;**Occlusion issues in crowds degrade the performance of detection and counting.**&zwnj; Occlusion is one of the key challenges in crowd counting, which requires an algorithm to actively adjust the viewpoint to mitigate this challenge. &zwnj;**To address this, the key component of the method, NLBN, is proposed to generate viewpoints according to the crowd distribution automatically.**&zwnj; It converts the difficult viewpoint selection problem into a relatively simple surface detection problem and generates viewpoints that reduce the degree of occlusion among the crowd. &zwnj;**Table 2 in the manuscript**&zwnj; shows that, when equipped with NLBN, ZECC significantly outperforms existing methods. &zwnj;**Table 1 in the Appendix**&zwnj; shows that NLBN reduces the obstruction rate of the line of sight, which contributes to the improved performance of ZECC.
>
> &zwnj;**A strategy utilizing height adjustment to improve exploration efficiency is proposed.**&zwnj; We explicitly consider the advantage of height adjustment in the exploration task. By using HAE, the agent obtains broad views, and irrelevant areas can be rapidly filtered out. This method mitigates redundant exploration caused by limited views at a fixed height and reduces the occlusion avoidance trajectory. &zwnj;**Figure 5 (a) in the manuscript**&zwnj; shows the efficiency improvement.
>
> &zwnj;**The system-level design serves as a foundation for future work.**&zwnj; ZECC not only serves as a baseline for further improvement but is also an automatic trajectory generation tool, which is important given the data scarcity in the embodied agent community [2]. Training-based methods can be developed based on ZECC to further improve performance.
>
> &zwnj;**A new task and dataset that supports this research.**&zwnj; ECCD is proposed, and it supports large-scale scenes and large object quantities simultaneously, enabling more complex algorithm designs.
>
>  > Q3: Can the authors provide insights or preliminary evidence on real-world applicability?
>
>   A3: We show a real application case in response &zwnj;**A1**&zwnj;. The challenge of real-world deployment lies in dataset generation and long-range UAV control. These challenges are being solved by the community gradually. [2] shows that 3D Gaussian splatting can be used to construct simulation scenes based on the real world, and [3] has provided possible solutions for real-world drone deployment in large-scale environments. We can use these solutions to extend this work.
>
> We still would like to emphasize that this work mainly focuses on solving the occlusion issue in crowd counting and improving exploration efficiency by using height information. We have demonstrated the effectiveness of the proposed methods in both simulation and a real-world application case. Sim-to-real techniques indeed are important to the community, and we are trying to improve these aspects in future works.
>
>  > Q4: How critical is the vision-language model (GPT-4V) to the success of ZECC?
>
>  A4: We conduct experiments on replacing GPT-4V with the open-source Qwen-VL-Max model and the tiny MiniGPT model. The results are shown in Table 2. It shows that the performance is consistent with different MLLMs. We believe this is because the images captured during HAE lack complex occlusion compared to LAE, making it easy to analyze, which results in consistent results for different MLLMs
>
> Table 2: Performance of different MLLMs.
>
>  | Method        | MAPE (%)     | TD (m)           |
> |---------------|--------------|------------------|
> | Qwen-VL-Max   | 17.67 ± 2.4  | 3752.79 ± 94.9   |
> | MiniGPT       | 17.58 ± 3.12 | 3809.09 ± 123.69 |
> | GPT-4V        | 18.71 ± 1.41 | 3722.45 ± 73.78  |
>
> [1] Wang, Jianyuan et al. “VGGT: Visual Geometry Grounded Transformer.” Computer Vision and Pattern Recognition (2025).
>
> [2] Gao, Yunpeng et al. “OpenFly: A Comprehensive Platform for Aerial Vision-Language Navigation.” (2025).
>
> [3] Lou, Jiabin et al. “Air-M: A Visual Reality Many-Agent Reinforcement Learning Platform for Large-Scale Aerial Unmanned System.” 2023 IEEE/RSJ International Conference on Intelligent Robots and Systems (IROS) (2023): 5598-5605.

---

> ### Author Response · Authors · 2025-08-05
>
> Dear Reviewer,
>
> Thank you again for your time and effort invested in the evaluation of our paper. We would like to ensure that we have addressed all your concerns satisfactorily. If there are any additional feedback you'd like us to consider, please let us know. Your insights are invaluable to us, and we're eager to address any remaining issues to improve our work.
>
> Thank you again for your time and effort in reviewing our paper.

---

### Decision · Program_Chairs · 2025-09-17

**Decision:**

Accept (poster)

**Comment:**

Initially the reviews were mixed but, after the rebuttal and discussion phase, they all recommend acceptance.